# Studying Biomolecular Protein Complexes via Origami and 3D-Printed Models

**DOI:** 10.3390/ijms25158271

**Published:** 2024-07-29

**Authors:** Hay Azulay, Inbar Benyunes, Gershon Elber, Nir Qvit

**Affiliations:** 1Independent Researcher, Koranit 2018100, Israel; ingral2@gmail.com; 2Faculty of Mechanical Engineering, Technion IIT, Haifa 3200003, Israel; b.inbar@campus.technion.ac.il; 3Faculty of Computer Science, Technion IIT, Haifa 3200003, Israel; gershon@cs.technion.ac.il; 4Faculty of Medicine in the Galilee, Bar-Ilan University, Safed 1311502, Israel

**Keywords:** protein, Bacterial microcompartments, origami, Kusudama, metamaterial, chiral, 3D-printing

## Abstract

Living organisms are constructed from proteins that assemble into biomolecular complexes, each with a unique shape and function. Our knowledge about the structure–activity relationship of these complexes is still limited, mainly because of their small size, complex structure, fast processes, and changing environment. Furthermore, the constraints of current microscopic tools and the difficulty in applying molecular dynamic simulations to capture the dynamic response of biomolecular complexes and long-term phenomena call for new supplementary tools and approaches that can help bridge this gap. In this paper, we present an approach to comparing biomolecular and origami hierarchical structures and apply it to comparing bacterial microcompartments (BMCs) with spiral-based origami models. Our first analysis compares proteins that assemble the BMC with an origami model called “flasher”, which is the unit cell of an assembled origami model. Then, the BMC structure is compared with the assembled origami model and based on the similarity, a physical scaled-up origami model, which is analogous to the BMC, is constructed. The origami model is translated into a computer-aided design model and manufactured via 3D-printing technology. Finite element analysis and physical experiments of the origami model and 3D-printed parts reveal trends in the mechanical response of the icosahedron, which is constructed from tiled-chiral elements. The chiral elements rotate as the icosahedron expands and we deduce that it allows the BMC to open gates for transmembrane passage of materials.

## 1. Introduction

It is difficult to ignore the fact that many biological structures can be described by geometrical mathematical functions [1], such as the Fibonacci series [2]. Still, there is much we do not know about how these structures self-organize in a hierarchical order from small building blocks to generate required properties, and how they dynamically change shape, in response to external stimulation. Limitations of current microscopic tools and the difficulties applying computer simulations to capture the dynamic response of biomolecular complexes and long-term phenomena call for new quantitative models and analytical tools to help increase our knowledge [3].

In a different direction, impressive advances have been made in origami research, which revealed that the art of folding paper into three-dimensional (3D) models is guided by mathematical and physical rules [4]. Combining the two, researchers have demonstrated that there are similarities between biological structures and origami. For example, origami folds were identified in the leaves and wings of insects that fold and unfold [5,6], origami folding principles were harnessed to study how proteins fold [7], DNA-origami methods were applied to construct 3D molecules [8,9], and particle origami-based robots that shrink and expand were designed to mimic the interaction between bimolecular components and complete collaborative tasks [10]. The above led to our previous research, in which we holistically compared protein and origami, from the creation process to the properties of the final structures [11].

The common approach to computer simulation of biomolecular structures is via the molecular dynamic (MD) method. The molecules are modeled along with the forces that support the structure, such as Van der Waals forces, salt bridges, hydrogen bonds, disulfide bonds, and the formation of a hydrophobic core. The physical dynamic response of atoms is calculated as the biomolecular structure changes toward an energetic equilibrium state [12].

However, the computational time and low accuracy of MD limit its applicability to small-scale molecular structures and short-term phenomena. Thus, a full atomistic molecular dynamics simulator to model conformation changes in protein-based membranes remains a great challenge [13]. The finite element analysis (FEA) framework, which is often applied to simulate the response of mechanical structures to forces, has been proposed as a supplementary tool to support the analysis of biomolecular structures [14]. It takes a continuum mechanics approach that does not detail chemical-biological features. Therefore, it appears attractive for tackling complex large domains and long-time phenomena at a relatively low computational cost. The FEA in this work is performed on models with shape and topology that is similar to the shape and topology of the biomolecular complex. Hence, the FEA can help understand how the high hierarchy structure would react to forces and how each building block in the tessellated structure conforms.

FEA requires as input a computer-aided design (CAD) model. However, with current commercial CAD software (e.g., SolidWorks© 2024), it is next to impossible to create tessellated models with many degrees of freedom. This limits our ability to analyze origami models with FEA simulation tools and the ability to learn about the mechanical properties of these complex structures. With the proper CAD software for modeling hierarchical tessellated structures, it will be possible to run FEA and 3D-print tangible models that can help validate in physical experiments the trends in the mechanical responses of the structures.

In this paper, we expand our comparison approach of proteins and origami models to complex structures and combine FEA, tessellated CAD modeling, origami models, 3D-printed models and experiments, and apply it to analyze the bacterial microcompartment (BMC). Based on the similarity to the biomolecular complex structure, we postulate how shell and vertex proteins of the icosahedron BMC may contribute to open gateways between the two sides of the BMC membrane in response to internal pressure.

## 2. Results and Discussion

The comparison approach is applied to a spiral-based assembled origami model to demonstrate the effectiveness of the presented approach. BMC is an icosahedron (a regular polyhedron with 20 faces) organelle composed of a proteinaceous shell structure that maintains a highly specialized environment for biochemical reactions inside [15]. Its shell is composed of multiple protein homologs that self-assemble to form the defined architecture shown in Figure 1A, and as a result BMCs are candidates for engineered molecular scaffolds, which can be used for different applications, such as drug delivery vehicles [16].

In this section we: (i) compare the structures of the main shell protein, which constructs the BMC, with origami models, (ii) compare the BMC structure with a Kusudama origami model, (iii) present CAD models that capture the shape of the origami model, (iv) present the resulting 3D-printed parts, and (v) present results from mechanical analysis and testing of the physical folded origami complex and 3D-printed parts, and explain how it may relate to the function that the BMC fulfills.

**Figure 1 ijms-25-08271-f001:**
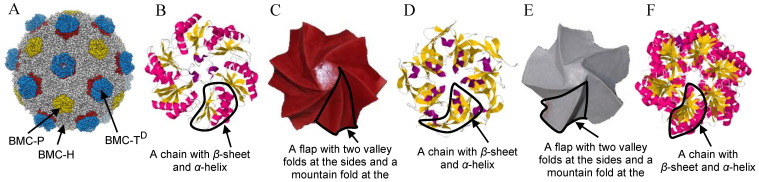
BMC proteins and origami models: (**A**) BMC structure (PDB: 6MZX), (**B**) top view cartoon model of a BMC-H protein, with cartoon marks for α-helices (red coils) and for β-strands (yellow flat shapes). (PBD: 3GFH). (**C**) origami flasher model with six flaps, (**D**) top view cartoon model of a BMC-P protein (PDB: 4N8F), (**E**) origami flasher model with five flaps, and (**F**) top view cartoon model of the BMC-T^D^ protein (PDB: 2A1B).

### 2.1. Level I—Compare the Shell Proteins with the Flasher Origami Model

#### 2.1.1. Structural Analysis

The BMC capsule is assembled from four main shell proteins: BMC-P, BMC-H, BMC-T^S^, and BMC-T^D^. BMC-P is a vertex protein, with five axisymmetric spiral chains (Figure 1D). BMC-H is the main building block that assembles the panels of the BMC (e.g., tiles), and it is a hexamer constructed from six axisymmetric spiral chains (Figure 1B) [17]. In this protein, every chain is formed by a *β*-sheet that is connected, on each of its sides, to a *α*-helix. The arrangement of axisymmetric units, constructed from *β*-sheet and *α*-helixes, appears in other protein complexes, such as CorA [18]. BMC-T^S^ has three axisymmetric chains. However, each chain is assembled from two similar sub-chains, and therefore has a hexamer axisymmetric structure, and BMC-T^D^, which is a double-stacked protein, where each stack is similar to BMC-T^S^ (Figure 1F). This protein has the same structure as the TRiC chaperon, which also has lower and upper anti-symmetric parts [18]. The BMC-T^D^ is located close to the center of the tessellated panel of the BMC, and it has a gated pore that changes from open to closed. The pores have been proposed to operate in an airlock fashion with opening and closure controlled by ligand binding, in a coordinated fashion across the shell surface [17].

#### 2.1.2. Mechanical Properties of the Flasher Origami Model

Spirals are universal patterns that appear on different scales from galaxies and hurricanes to bio-structures, and researchers are still trying to understand what makes them unique and how this shape relates to their function. For example, examining the relationship between vortical turbulence defect motions of wave propagation, Rho-GTP signaling proteins on the membrane of starfish egg cells, cytoskeletal remodeling, and cell proliferation [19]. BMC shell proteins are also spiral and that raises the question of what structure–activity relationship the pattern fulfills at the local and macro levels.

The flasher origami model has a spiral-like fold that opens and closes in a circular motion (Figure 1C and Figure 1E, respectively). It comprises axisymmetric twist triangular flaps, and similar to the chain structure in the BMC-H and BMC-P proteins, the flaps in the flasher are constructed from valley folds on the sides and a mountain fold in the center. The folds in the valley are shared by two neighboring flaps, so folding one flap results in a synchronized fold of the whole flasher.

In a previous paper [11], we showed that the structure of the mechanosensitive ion channel protein (MscS), which is a membrane protein with an axisymmetric structure, similar to the membrane protein of the BMC, is analogous to the flasher origami model. Mechanical analysis of the flasher model demonstrated how the MscS protein can increase/decrease its length simultaneously along axes perpendicular to an axial stretching force. Structures with such properties are known as “auxetic”, which means that, counter to typical natural behavior and intuition, it expand in 3D when stretched, and contract in 3D when compressed [11]. Since the structure shrinks and expands in rotational motion, it is defined as “chiral”. The definition here for chirality refers to the mechanical structural property and not to the chemical chirality which results from the orientation of atoms in a stereoisomer. Furthermore, when the folds tilt vertically, the structure can rise from a flat shape to a dome shape via radial and tangential forces (Figure 2) [20].

The origami flasher model and the BMC-H and BMC-P membrane proteins demonstrate a similar rotational chiral structure. Hence, these proteins should be able to simultaneously expand/contract in a synchronized motion along the axes, meaning their domains stretch up from the base during contraction. Based on several papers [15,17], the proteins that construct the BMC are considered to have an axisymmetric rotational structure, which due to similarity to the flasher origami model is also chiral.

### 2.2. Level II—Comparing the Escherichia coli BMC with an Origami Tessellation Model and a 3D-Printed Part

#### 2.2.1. The BMC Structure

The BMC shell is only about 30 to 40 Å, which approximately corresponds to the width of a single layer of shell proteins. The shell proteins form a tightly packed two-dimensional (2D) lattice that functions as a selectively permeable membrane. The shell structure resembles the spacing found in 2D crystal arrays, suggesting they provide a structural basis for a required function (Figure 3A). Hence, it has been suggested that the almost seamless 2D packing has physiological relevance as it could provide an efficient barrier to prevent leakage of toxic by-products into the cytoplasm. To understand how the multi-scale icosahedrons’ chiral structure serves the function of the BMC, we need to record the dynamic response of the compartment, as well as the motion of individual proteins. However, currently, microscope resolution limits the ability to follow the dynamic motion of biomolecular structures and the real-time response of the proteins that construct them. In addition, the interactions among shell proteins and their arrangements within facets, edges, and vertices remain elusive. To help bridge this gap, we search for analog origami and 3D-printed models that can help analyze the mechanical response of the BMC and its structure–function relations.

#### 2.2.2. The Spiral-Based Origami Model

It is possible to construct icosahedrons and other geometrical polyhedrons using origami [21]. The first approach, in which a single paper is folded with a repeated pattern and then folded to construct the polygon, belongs to the tessellations branch of origami (Figure 3A). The second approach, which belongs to the modular origami branch, is to assemble a desired structure from multiple folded building blocks [22] (Figure 3B,C). Interestingly, structures with patterns similar to origami tessellations can be found in architecture metamaterials, which are engineered structures that achieve desired macro properties via controlled micro-patterns [23,24,25], which can be used in different applications, e.g., implants [26].

**Figure 3 ijms-25-08271-f003:**
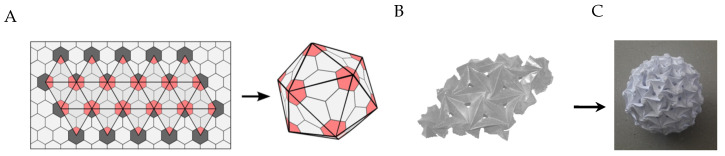
Folding an origami icosahedron: (**A**) Folding triangular planes on a sheet into an icosahedron (the figure is reprinted and revised from reference [27] under the terms of the Creative Commons Attribution 4.0 International License), (**B**) A tessellation of six-flap spiral folded flashers, and (**C**) An icosahedron that is constructed from assembly of origami flashers.

By combining six- and five-flap flasher origami models, which are analogs with the six- and five-chain BMC-H and BMC-P proteins, we constructed a Kusudama tessellation model that mimics the 3D structure of the BMC (Figure 4). Six-flap flashers form triangular faces, and the faces meet as a five-flap flasher at angles that impose the 3D structure of an icosahedron. The result is a tiled pattern of six- and five-flap flashers, which is similar to the tiled pattern of proteins in the BMC with pentamer proteins at the corners.

In tessellation models with regular tiling, which are constructed with a single type of tile, it is possible to cover the surface plane only with unit cells with a ligament number of n = 3, 4, and 6, namely triangles, squares and hexagons. Therefore, keeping rotational symmetry and connecting each ligament in a unit cell to the ligament of the adjacent unit cell, the hexamer BMC-H proteins can tile a plane, while the pentamer BMC-P protein cannot [28]. Combining only BMC-H and BMC-P proteins it is no longer possible to cover a plane while maintaining rotational chirality. A mix-tiled structure with BMC-H and BMC-P proteins only exists when the BMC-P proteins tilt relative to the plane, creating a 3D structure, in which every ligament in the BMC-P is connected to a BMC-H protein located on a different plane (Figure 4). Bacteria can store enzymes and other proteins in polyhedral 3D compartments by combining BMC-H and BMC-P proteins. The difference between the five- and six-chain shell protein results in specific structural properties [29], and studies demonstrated the intrinsic potential of shell protein to assemble autonomously into a bacterial microcompartment in a Lego-like assembly principle [15].

The origami Kusudama model is constructed from six- and five-flap flashers, where the flaps of every two adjacent flashers are glued. Hence, the icosahedron planes, which are constructed from the six-flap flashers, have a rotational symmetry of six, and they are auxetic as all the flashers shrink/expand synchronically [30].

To try to understand what the mechanical properties of the microcompartment structure are, we conducted a simple experiment and recorded the response of an icosahedron origami model, which is made from six-flap origami flashers that tile the planes and five-flap flashers at the corners. The structure was inflated using a balloon that was inserted into the inner cavity (Figure 5), and as it expanded, the diameter of the origami model more than doubled its length.

#### 2.2.3. Finite Element Analysis (FEA)

FEA is an efficient way of analyzing different structural variants. To allow cost-effective analysis, we created V-rep models of the origami complex. The CAD models combine chiral tiles that are tessellated to assemble variants of icosahedrons with different levels of roundness at the vertexes from icosahedrons to a sphere (Figure 6A,B). Important to note that the CAD model represents continuum material, and it does not directly represent each of the bonds and chemical interactions that construct biomolecular structures. However, once there are origami models and mechanical mechanisms that are similar to the shape of a protein or a biomolecular complex, we can analyze them to try and learn about the trends of how such structures can contribute to the function that the biomolecular complex fulfills.

The dimensions of the CAD model and the BMC are specified in Table 1. Initial simulations indicated that the outer shape of the structure is of importance to its mechanical response to internal pressure. Therefore, we analyzed the response of different structural variants between an icosahedron and a sphere (Figure 7). In the analysis, a pressure force was applied on the internal planes of the tiles, and a five-chain vertex tile was constrained by a fixed support.

The mechanical changes due to forces are derived from an FEA in the commercial ANSYS^®^ software (Ansys^®^ Mechanical Products, Release 2021R) [32]. Based on the response to an internal pressure of icosahedrons with different levels of roundness, it appears that in a polygonal icosahedron with vertexes that are not rounded (Figure 7(A1–A3)), the planes expand locally, and the vertexes remain stiff. Hence, the BMC structure does not expand uniformly, like a balloon. As the vertexes in the CAD model are rounded and the structure becomes a semi-icosahedron (Figure 7(B1–B3)), the stiffening effect of the vertexes diminishes. Finally, when the BMC is a sphere (Figure 7(C1–C3)), the structure expands uniformly.

#### 2.2.4. 3D-Printed Icosahedron

To verify that the simulation results of the CAD model are similar to the response of a physical model, we 3D-printed an icosahedron and sphere parts we can test. The models are constructed from chiral tiles that have six chains at the surface and five chains at the vertices (Figure 8).

The first models were 3D-printed on a Stratasys^®^ J55 printer, applying several materials for the vertex tiles and plane tiles (Figure 8). However, since these materials easily deform, other models were printed with a Multi-Jet-Fusion HP^®^ printer using PA 12 material (Figure 9).

In the experiment, an icosahedron part was inflated using a balloon that was inserted into the inner cavity through a gap that was created when we removed one vertex tile (Figure 8A). The inflation was rapid; however, it is possible to notice the rotation of the chiral elements in the planes when the structure expands (Figure 9). Furthermore, looking at the silhouette of the icosahedron one can notice that the faces expand, while the chiral elements at the corners remain stiff. Hence, the results from experiments (Figure 9) and simulations (Figure 9(A1–A3)), of an icosahedron that expands via internal pressure, are in good agreement.

## 3. Materials and Methods

### 3.1. Comparison Approach

We developed a comparison approach for biomolecular complexes and assembled origami models, based on a previously developed method for single- and multi-domain proteins [33]. The comparison is applied in a hierarchical order, from the building blocks to the complex structure (Figure 10). This helps create a scaled-up origami model and 3D-printed parts to analyze the structure–function relationship.

The approach addresses details such as: (i) shape, (ii) structure (iii) mechanical properties, and (iv) arrangement of building blocks in the bio-structure. The analysis workflow is presented in Figure 11:(1)Selecting and dividing—Select a biomolecular complex and divide it into the lower-level proteins that assemble the structure.(2)Identifying—Identify in the literature [34], or design original origami models [35], with shape/conformation similar to lower-level protein structures.(3)Comparing—Level I—Analyze and compare the shape and structure of all the individual proteins that were identified in step 2 with origami models. This step applies the comparison approach presented previously [33] and applies knowledge from mathematics, physics, and origami-based research.(4)Assembling—Assemble the individual folded models into an origami complex. Origami uses different approaches to creating a model that combines many folded sub-units. The tessellation branch is based on folding a single continuous paper with a repeated folded pattern, and modular branches that connect individual folded units into a higher hierarchy complex model.(5)Comparing—Level II—Analyze and compare the shape and structure of the complex bimolecular structure from Step 1 with the assembled origami model that combines the individual origami models from Step 4. The comparison approach applied during analysis is similar to the steps described in Step 3. The origami model can be translated into a CAD model that is further analyzed via FEA.(6)Creating physical models—Creating physical models from folded paper and translating the CAD model to 3D-printed models that mimic the mechanical trends of the structure of the origami model and capture the shape and structure of the biomolecular structure.(7)Conducting experiments—With the origami and 3D-printed models from step 6 it is possible to conduct experiments to learn about the integral properties of the hierarchical biomolecular complex structure. Analyzing the 3D-printed model via FEA and testing the physical 3D-printed part and the complex origami model, it is possible to demonstrate how an overall structure reacts to forces, and what could be the motion of individual unit cells.

### 3.2. The Origami Model

Origami is a constantly evolving field and there are various sources for folding models, from basic known folding patterns to new designs with thousands of folding steps. In addition, there are several dedicated design tools to create computerized foldable origami models [35,36]. Origami is classified into different types based on various parameters, such as origin (e.g., traditional Japanese origami) technique (e.g., kirigami, crumpling and wet folding), properties (e.g., action and pop-up origami) and objective (e.g., jewelry, and tea bag folding).

In this paper we create the origami Kusudama model, which is a branch of modular origami that describes ball-like models with repeating folded units [37]. The model is constructed from individual unit cells that are folded individually and compared with the individual proteins in the complex biomolecular structure. Then, the individual units are connected to create an origami model with a tessellated hierarchical structure.

### 3.3. Finite Element Analysis (FEA)

FEA is the cornerstone of mechanical simulations, and it is widely applied to analyze the reaction of structures, which are made from continuum materials such as metals and polymers. FEA is also a tool for analyzing the properties of biological structures, which are generally on a very different size scale from mechanical structures. Although it does not consider the exact physical and chemical phenomena that govern interaction in such structures, we assume that the similarity of the shape of the biomolecular structure and the origami model is not a coincidence, and that it plays a role in the function it fulfills. Therefore, performing this analysis on a CAD model that represents the analog origami model can teach us the trends that such structures follow when subjected to forces. An advantage of the CAD and FEA model is that we can apply small changes to the model and explore how this affects the mechanical properties of the structure. Tiled structures, with repeating patterns, are a special and important case. Tessellated structures and specific shapes and topologies of tiles were widely studied, in the context of mathematics, origami and metamaterials, and we know to correlate some of them with specific mechanical properties. For example, it has been shown that the elasticity of a tiled structure increases as the number of ligaments in the chiral unit cell decreases [38]. In another study, numerical simulations have been applied to study the mutual affect on shape and orientation of closed membrane and its membrane vesicles and chiral nematic rods when the membrane deforms [39].

### 3.4. The CAD Model

Designing a tiled model for simulations using conventional CAD tools is a complex task, as it is not adjusted for modeling a structure with a defined overall shape and volume, from a combination of different chiral tiles that exactly fit together. Furthermore, due to the details in the hierarchical model, which incorporates many rounded features, the size of such a data file will be large and preparing the model (meshing) for FE analysis will increase the file size exponentially, making it too complex to handle using standard computers.

To overcome the above challenge, and to allow cost-effective analysis, we chose to directly create a mesh CAD model that captures the properties of the folded origami model. The model was created using CAD software that exploits volumetric representations (V-reps) and is developed at the Technion IIT. The software allows designers control over the features of each tile at any location in space [40], and modeling of complex lattice designs as well as ease migration to analysis tools, properties we draw upon herein. Further, V-reps also allow one to represent (graded) heterogeneity in (the material properties of) the interior, a property we expect to exploit in the future. Yet, the modeling process using V-reps is similar to traditional boundary representation (B-rep)-based CAD systems—building freeform (spline) curves, extending the curves to surfaces, via constructors such as extrusion, ruling, or sweep, only now to also expand into volumetric trivariate spline functions, in a way that is transparent to the end user.

### 3.5. The 3D-Printed Model

Additive manufacturing (AM) is a disruptive technology which allows you to 3D-print parts that cannot be manufactured with other technologies [41]. This is true for tiled lattice structures, which is quickly becoming a new multidisciplinary research field that focuses on how to design hierarchical structures for desired properties.

A 3D printer generally translates a CAD model to a printed part. To comply with the (STL) file format that many 3D-printers require, the CAD models in our work were converted from V-reps trivariate elements into surfaces, converted again into polygons and saved as files in the OBJ file format (preserving their colors), only to be loaded into the GrabCAD GUI of the printers of Stratasys^®^. 3D-printed on a Stratasys^®^ J55 printer (Eden Prairie, MN, USA), using their Vero colored materials (see https://www.stratasys.com/en/3d-printers/printer-catalog/polyjet/j55-prime, accessed on 5 April 2024) and with a Multi-Jet-Fusion HP^®^ printer (Palo Alto, CA, USA) using PA 12 material.

### 3.6. Experiments

We conducted experiments to test the trends of how origami and 3D-printed parts respond to pressure. In the experiments, a balloon was inserted into the inner cavity of the parts and then inflated. The objective was to analyze the correlation between the motions of the complex structure and the individual tiles. The experiments provide a physical complementary tool that validates the FEA results.

## 4. Conclusions

A two-level comparison approach, which incorporates simulations as well as testing of physical models and parts, is proposed here to analyze the mechanical properties of biomolecular complexes. The approach is demonstrated in the comparison of the biomolecular BMC and the Kusudama flasher-based origami model, which are both hierarchical structures assembled from building blocks. The origami models that are analogous to the BMC shell proteins are auxetic-chiral, which means that pressure on the BMC is translated to expansion and contraction of these proteins through rotational motion. The origami tessellation model and a 3D-printed part that captures its shape help to explain how the structure of the micro-level shell and vertex protein contribute to the dynamic motion of the higher-level BMC protein structure, and how the BMC is assembled to expand and shrink. Based on the mechanical response of the icosahedron shell in simulations, the structure of the membrane protein with a pore at its center, and the location of the BMC-T^D^ at the center of the icosahedron plane, it appears that in the case of internal pressure, the BMC-T^D^ proteins will experience high mechanical stretch forces. These forces would lead to a large rotation of the protein chains, compared to the rotation of proteins that are closer to the edges and vertexes of the icosahedron planes, and at a certain rotation angle, the outer and inner openings of the BMC-T^D^ pore should be large enough to allow high transition between the internal and external sides of the microcompartment. Hence, we postulate that the structure of the icosahedrons in combination with the structure of shell proteins fulfill a biological function, i.e., some shell proteins act as valves that transport molecules across the microcompartment.

The comparison process also led to the design of icosahedron- and sphere-based structures that are constructed from chiral tiles. These structures have unique properties that may help achieve desired functions in future engineering applications, such as controlled expansion and filtering. Following the abovementioned challenges in designing the structural features of biomolecular complexes with hierarchical tiles in CAD models, and analyzing their properties in FEA, it appears that the approach and computational tools presented in this paper can open the door to new opportunities in the design of structures with desired properties.

The continuum material-based CAD and FEA models cannot directly represent each of the bonds and chemical interactions that construct biomolecular structures. However, with origami models that are similar by shape and structure to proteins and biomolecular complexes, it is possible that models and analysis approaches from the mechanical engineering discipline can support additional approaches to try and learn more about the functions that the biomolecular complex fulfills and guide the design of de novo proteins and nanomaterials with desired properties.

In future work, to further bridge the gap between biochemical and origami and 3D-printed models, we would like to apply a hybrid simulation approach combining MD simulations for low-complexity proteins and FEA for high-complexity hierarchy protein-based structures.

### Significance Statement

To our knowledge, this study is the first to compare complex hierarchical structures in biomolecules with origami and 3D-printed models. Origami and 3D-printed models that are analogous to biomolecular complexes can offer insights into how the dynamic movements of individual proteins contribute to the properties and functions of the combined biological structure. The proposed comparison process can also help translate the structure of biomolecular complexes into new hierarchical metamaterials.

## Figures and Tables

**Figure 2 ijms-25-08271-f002:**
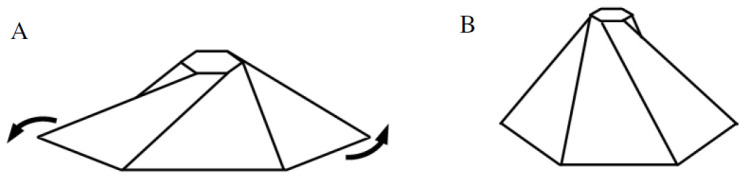
Dome top changes as the structure rotates: (**A**) top open, and (**B**) top closes after rotation.

**Figure 4 ijms-25-08271-f004:**
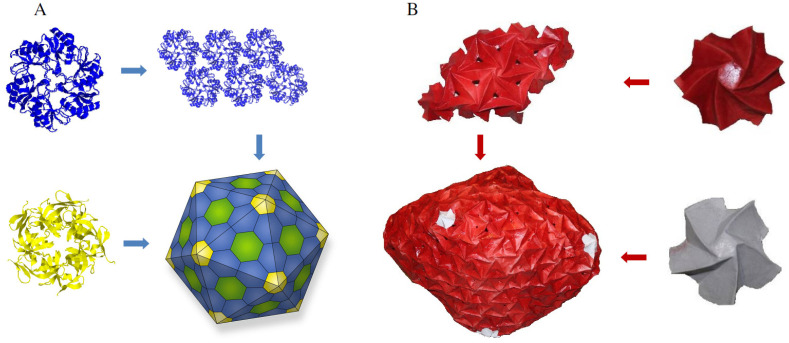
The BMC and the origami tessellation model: (**A**) The BMC structure with the hexameric BMC-H (blue), which is the main unit cell that constructs the panels, the hexameric BMC-T^D^ (green), which is located at the center of the planes and the pentameric BMC-P (yellow), which is located at the vertices (the figure is reprinted and revised from reference [17] under the terms of the Creative Commons Attribution 4.0 International License), and (**B**) an origami-based compartment, with a hexameric flasher as the main unit cell, and a pentameric flasher at the corners.

**Figure 5 ijms-25-08271-f005:**
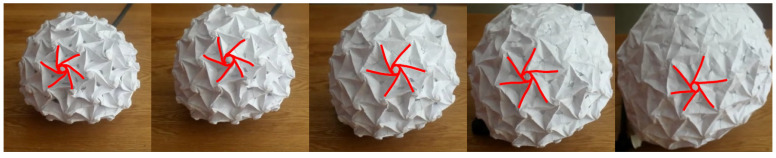
Dynamic experiment with icosahedron origami models—inflating the structure with air. The folding lines of a flasher that are marked in red show how it changes as the model expands.

**Figure 6 ijms-25-08271-f006:**
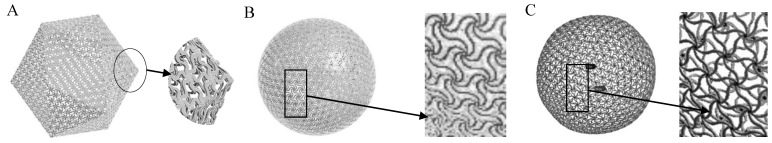
FEA results—deformation of (**A**) an icosahedron with elements with six ligaments that tile the planes, and elements with five ligaments at the corners. (**B**) Sphere with elements with six ligaments that tile the planes and elements with five ligaments at the corners. (**C**) FEA results—deformation of the elements due to internal pressure.

**Figure 7 ijms-25-08271-f007:**
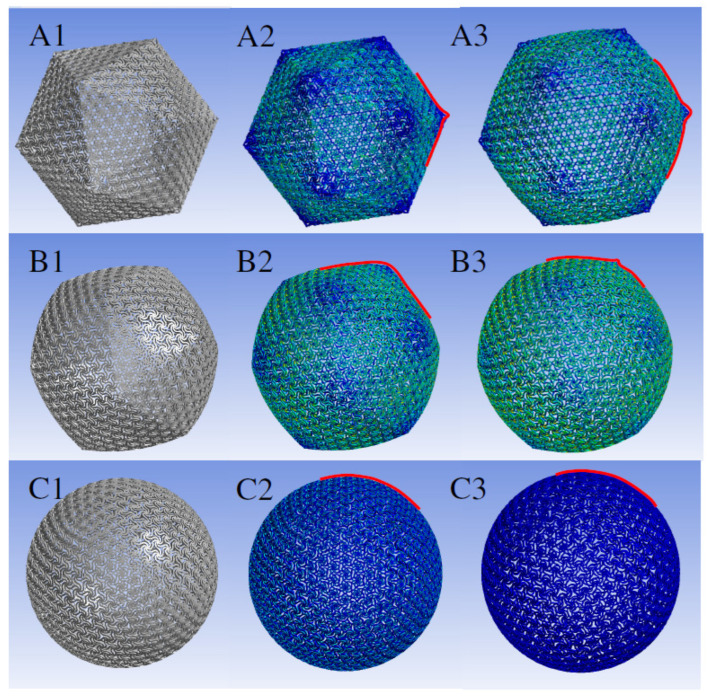
From an icosahedron to a sphere—(left column)—structures with different curvatures, from an icosahedron (**A1**) to a sphere (**C1**). In the FEA simulations, a vertex tile is fixed, and internal pressure is applied to the tiles. The FEA deformation results of the models are shown in the center and right columns (**A2**–**C3**). The deformation of the silhouette is marked by a red line that follows the plane and the corners. The color map represents the relative deformation of the elements due to the applied force, where green represents deformation and blue no deformation.

**Figure 8 ijms-25-08271-f008:**
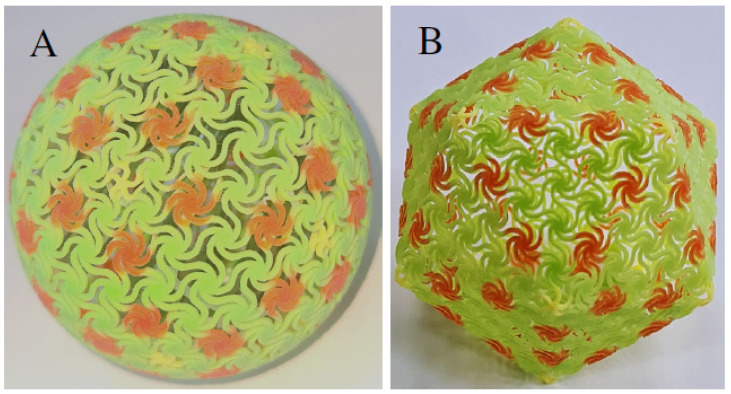
Parts with chiral elements that were 3D-printed with Stratasys^®^ J55: (**A**) Sphere. (**B**) Icosahedron.

**Figure 9 ijms-25-08271-f009:**
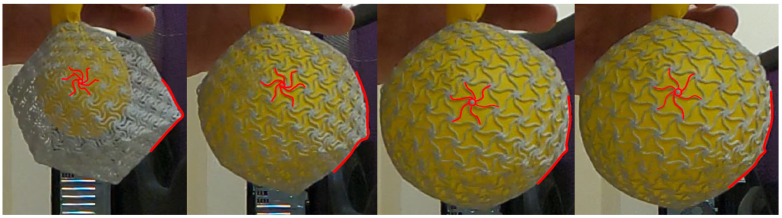
An experiment with a 3D-printed icosahedron. A chiral element and part of the silhouette are marked in red to demonstrate how the structure changes during expansion (from left to right).

**Figure 10 ijms-25-08271-f010:**
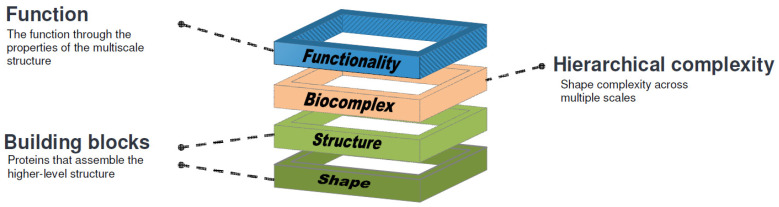
The hierarchical bottom-up arrangement of a layered bio-structure, from building blocks to function.

**Figure 11 ijms-25-08271-f011:**
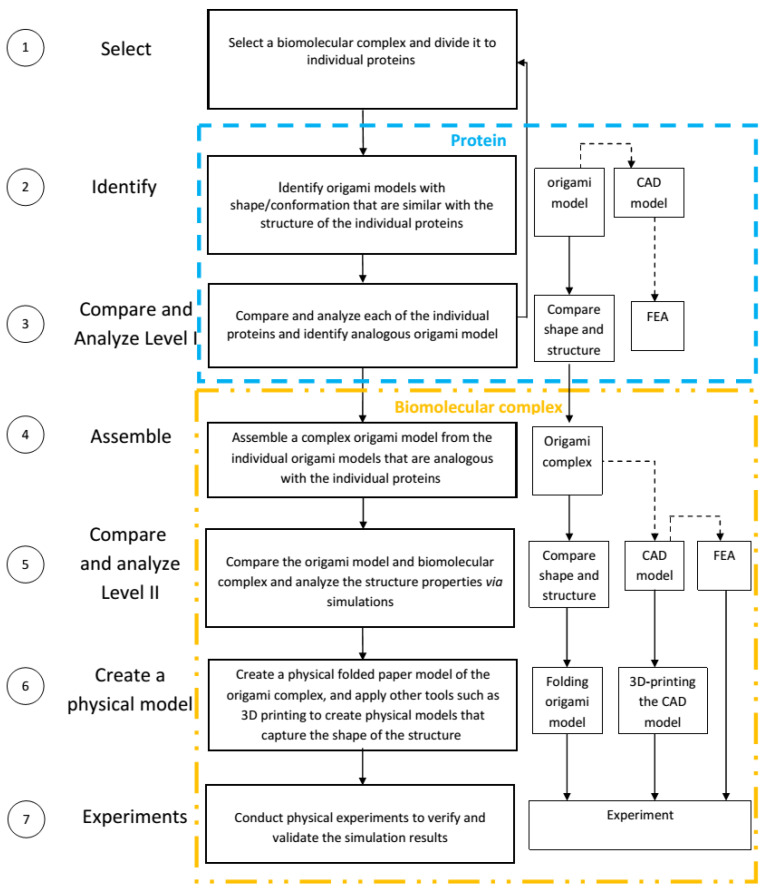
A workflow of the approach to comparing bimolecular complexes and origami models.

**Table 1 ijms-25-08271-t001:** The dimensions of the different models.

Model	Diameter
BMC [31]	~10,000 Å
Origami icosahedron	~205 mm
CAD/FEA model	65 mm
3D-printed icosahedron	65 mm

## Data Availability

The original contributions presented in the study are included in the article, further inquiries can be directed to the corresponding author.

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
