# Peer review of "Studying Biomolecular Protein Complexes via Origami and 3D-Printed Models"

_ijms, 2024, doi:10.3390/ijms25158271_

Round 1

Reviewer 1 Report

Comments and Suggestions for Authors

In this manuscript the authors presented a work on the comparison of origami and biomolecular structures. It is not clear which is the novelty of the manuscript. Moreover, all the sections of the manuscript lack of scientific rigour. The description of the experimental part is only sketchy and the results are not adequately supported. The following are some advice for the authors:

·       The introduction section is very limited. Please expand this section better analysing other studies in this field.

·       The structure of the manuscript is not clear. A materials and methods as well as a result section is missing.

·       Finite element analysis is only cited and no results nor how boundary and domain conditions were set are reported in the manuscript.

·       It is not clear which is the reason of using the 3D printed model compared to biomolecular complexes. Moreover, no detail is reported about the 3D printing process. Which technology was used? Parameters? How was the balloon inserted? Which is the dimension?

·       English language should be improved. Please carefully revise the language throughout the manuscript.

Minor comments:

·      Line 256, please correct “metamaterial”

·      Check paragraph number “3.2.43”

Reviewer 2 Report

Comments and Suggestions for Authors

The manuscript “Studying Biomolecular Protein Complexes via Origami and 3D-Printed Models“ [ijms- 3051444 -v1] written by Hay Azulay, Inbar Benyunes, Gershon Elber and Nir Qvit describes an investigation on biomolecular protein complexes, likely present in bacterial microcompartments. The authors use self made Kusudama flasher-based origami models for this investigation.

The reviewer has expertise in the molecular field of organic chemistry and structure determination and hence mostly refers from this point of view within the review of the manuscript.

The aim of describing complex relationships in protein chemistry using models that can be self created is an interesting and worthwhile goal for the reviewer. However, it should be taken into account that models are always only an imperfect representation of natural structures and the associated dynamics. Models are therefore simplifications that are intended to contribute to a better understanding of natural processes. However, one should always be aware that there are limits here and that not all natural processes can be explained with one and the same understandable model. (One example is the vector model by F. Bloch, which makes the basics of NMR understandable; however, this reaches its limits when describing e.g. two-dimensional NMR spectra, as the physical principles used are not implemented in this model.)

The reviewer therefore welcomes the model presented as a clear representation of the basics of BMCs. This is certainly an aspect of the presentation of models that should not be neglected and that is worth publishing in an appropriate form in a scientific journal.
However, the reviewer notices some approaches by the authors that go beyond this and that the authors themselves should examine very critically. The point here is that conclusions are drawn from the behavior of the macroscopic model about the microscopic and molecular world of BMCs. From the reviewers' point of view, the authors are crossing a line here as to what conclusions can be drawn from models in general and in the specific case of origami and 3D-printed models.

Therefore, the manuscript is not in a form to be published in “International Journal of Molecular Science“.

However, the manuscript is of some interest in the fields of modelling, protein chemistry and to some extend in chemical education. The authors are therefore encouraged to take the fundamental criticism described above into account and to fundamentally revise the manuscript.
It would probably be more sensible to publish it in a revised form in a journal that focuses more on (high level) chemical education. When revising, authors should take the aspects mentioned below into account.

a) The authors argue intensively with the axial chirality of the protein pentamers and hexamers. Here they show (in Fig. 4) that the model allows racemization by inversion of the atropisomers. The reviewer is not aware of any protein for which this has ever been described. The molecular basis for this is not given (see b and c). The authors are therefore encouraged to bring additional expertise from experts in the molecular field into their team and thus cover this aspect (and other molecular aspects) completely.

b) In some passages the authors refer to publications on complex dynamic relationships, which are then transferred incorrectly. For example reference 20 describes different processes than those shown by the authors in their model.

c) The authors should consider more intensively the formation and repulsion forces that may occur in proteins (molecular structures). These are completely different from the forces in their 3D model. Particularly with regard to possible geometric expansion.

d) Experimental results should be incorporated much more in the seven-step workflow (Fig 2) and not just at the end. Known experimental data and self-generated results can be introduced at several points earlier and weaknesses in the model and the conclusions drawn from mistakes in interpretation can be eliminated.

e) The geometry of the protein under consideration should be taken into account more closely. Not all pentamers have a chalice structure (Fig. 6).

f) The reviewer would also be interested in more details on how the models are made. This would enable them to be refined and used by other researchers. This would potentially enable further use.

Reviewer 3 Report

Comments and Suggestions for Authors

In this work the authors studied the biomolecular protein complexes via origami and 3D printed models. The idea of this work is interesting. However, the reviewer believes that this work lacks fundamental insights into the correlation between the protein and 3D models, since they are in very different scales. Much more physical insights should be provided and the simulation analysis needed to be strengthen.

It is known that protein complexes maintain their structure due to many types of non-covalent interactions. These interactions cannot be well described by a solid 3D print system. With only covalent bonds the amino acids cannot fold into a well defined structure. The reaction of a protein to the external stimuli such as pressure is not trivial to be described by the CAD simulations.

The most precise tool regarding the complex structure would be molecular dynamics simulations, there one would be able to understand what happens to the complex, by utilizing unbiased or biased technique, where the atoms interactions are well defined.

Therefore I cannot recommend this work to be published, because I believe apparently, methods in very different scales are used to study nano-scale properties. Although I believe this is an interesting try.

Comments on the Quality of English Language

Minor editing of English language required

Round 2

Reviewer 1 Report

Comments and Suggestions for Authors

the paper was revised as required

Reviewer 2 Report

Comments and Suggestions for Authors

The manuscript now entitled “Studying Biomolecular Protein Complexes via Origami and 3D-Printed Models“ [ijms-3051444-v1] written by Hay Azulay, Inbar Benyunes, Gershon Elber and Nir Qvit has been revised by the authors. The reviewer is grateful to the authors for taking to heart the previous comments, also from other reviewers and for addressing several concerns.  

A detailed list with answers of the authors to all reviewer’s comments, point by point, is added to the new submission. The authors have taken all of the comments of the reviewer as well as obviously of other the reviewers into account. The authors have answered to the concerns and made some comprehensive respective corrections, additions, and changes in the newly submitted manuscript. All changes in the manuscript make sense. They have been carried out with direct reference to the comments. In particular, the reviewer's concerns regarding possible conclusions from models about natural phenomena have been taken into account by the authors. The corresponding explanations in the text appear to be conclusive to the reviewer.

Hence, the quality and readability of the manuscript have been improved in the newly submitted version. The manuscript now fully describes possibilities and limitations of self made Kusudama flasher-based origami models to describe biomolecular protein complexes, which are likely present in bacterial microcompartments.
From the reviewer's perspective the manuscript is therefore acceptable for publication in the “International Journal of Molecular Science”.

Reviewer 3 Report

Comments and Suggestions for Authors

I believe this work can be accepted.